# What Inhibits Natural Killers’ Performance in Tumour

**DOI:** 10.3390/ijms23137030

**Published:** 2022-06-24

**Authors:** Ines Papak, Elżbieta Chruściel, Katarzyna Dziubek, Małgorzata Kurkowiak, Zuzanna Urban-Wójciuk, Tomasz Marjański, Witold Rzyman, Natalia Marek-Trzonkowska

**Affiliations:** 1International Centre for Cancer Vaccine Science, University of Gdansk, Ul. Kładki 24, 80-822 Gdansk, Poland; ines.papak@phdstud.ug.edu.pl (I.P.); elzbieta.chrusciel@ug.edu.pl (E.C.); katarzyna.dziubek@phdstud.ug.edu.pl (K.D.); malgorzata.kurkowiak@ug.edu.pl (M.K.); zuzanna.urban-wojciuk@ug.edu.pl (Z.U.-W.); 2Department of Thoracic Surgery, Medical University of Gdansk, 80-210 Gdansk, Poland; tomasz.marjanski@gumed.edu.pl (T.M.); wrzyman@gumed.edu.pl (W.R.); 3Laboratory of Immunoregulation and Cellular Therapies, Department of Family Medicine, Medical University of Gdansk, 80-210 Gdansk, Poland

**Keywords:** natural killer cells (NK), tumour microenvironment (TME), activation receptors, inhibitory receptors, immunosuppression

## Abstract

Natural killer cells are innate lymphocytes with the ability to lyse tumour cells depending on the balance of their activating and inhibiting receptors. Growing numbers of clinical trials show promising results of NK cell-based immunotherapies. Unlike T cells, NK cells can lyse tumour cells independent of antigen presentation, based simply on their activation and inhibition receptors. Various strategies to improve NK cell-based therapies are being developed, all with one goal: to shift the balance to activation. In this review, we discuss the current understanding of ways NK cells can lyse tumour cells and all the inhibitory signals stopping their cytotoxic potential.

## 1. Introduction

Natural killer (NK) cells are known as large granular lymphocytes of the innate immune system capable of detecting and lysing tumour cells. The role of NK cells in anti-cancer response is not limited to tumour lysis, but also involves activation of the adaptive immune system [1].

The success of NK cell-based immunotherapies in the last 30 years has been greatly attributed to advancements in the understanding of NK cell biology. Several promising results in laboratory settings have been already translated into phase I and II clinical trials [2,3]. However, in many oncologic patients, NK cell responses are compromised, either due to premature immunosenescence or the immunosuppressive effect of the tumour microenvironment. The exact mechanisms that can be used to enhance and maintain NK cell function in vivo still need to be elucidated. With this review, we provide an overview of inhibitory factors responsible for NK cells suppression in oncologic patients. Technological progress during the last few years resulted in generation of genetically modified CAR-NK cells [4]. However, there is still no efficient strategy that would protect these trained killers from the cancer immunosuppressive network. We believe that the problem of the inhibitory environment of the tumour should be addressed as early as in in vitro studies. We hope that this paper will shed light on various mechanisms of NK cell suppression caused by the tumour and contribute to development of novel combinatory therapies that will improve clinical outcome in patients with malignancies.

### 1.1. Predominant Subsets of NK Cells in the Blood

Natural killer (NK) cells are large granular innate lymphocytes that represent 5–20% of total human blood lymphocytes. They are involved in the defence against various microbial infections and autoimmunity, but also tolerance, implying NK heterogenicity. NK cells defend the body against viral infections and participate in the immunity against various malignancies, with eliminating metastatic tumour cells being one of the most important features. For this reason, NK cells are of great interest when it comes to designing novel tumour immunotherapy [5,6].

NK cells from human blood are identified based on the cell-surface density of the CD56 adhesion molecule and Fc-gamma receptor CD16 (FcγRIIIa) involved in antibody-dependent cellular cytotoxicity (ADCC). CD56dim population dominates in peripheral blood and has high expression of Killer cell Ig-like receptors (KIR). They also produce high levels of cytotoxic effector proteins (e.g., perforin and granzymes) and CD16 receptors, and consequently have a high potential for direct tumour lysis. As the most potent cytotoxic part of innate immunity, this subset is often compared to cytotoxic CD8+ T-cells [7].

The second most numerous NK cell population in human blood is identified using the CD56bright t CD16 phenotype. This subpopulation has low KIR expression and high expression of the NKG2A/CD94 inhibitory receptor. It also secretes large amounts of T helper cell type 1 (Th1) cytokines such as interferon-gamma (IFNγ), interleukin 10 (IL-10), interleukin 13 (IL-13), tumour necrosis factor alfa (TNFa), granulocyte-macrophage colony-stimulating factor (GM-CSF), and various chemokines (CCL1, CCL2, CCL3, CCL4, CCL5, and CXCL8) [8,9]. IL-10 and IL-13 also regulate various aspects of allergy-induced inflammation, mediating T helper cell type 2 (Th2) immune responses [10,11]. Such abundant and diverse cytokine production suggests that this subset is involved in early immune system activation, as well as modulation of the various adaptive immune system responses.

At this point, three additional NK cell subsets can be identified: CD56dimCD16-, CD56bright tCD16dim, and CD56-CD16+; the last of which is the only NK cell subset negative for CD56 and has been reported to expand in chronic HIV infection, malaria, endemic Burkitt lymphoma, and prolonged SARS-CoV-2 infection [9,12,13]. This population characterises high expression of KIRs, NKG2C, NKp30, 2B4, CD57, and TRAIL [12].

When it comes to murine NK cells, it is important to note that they do not express a murine homolog of CD56. Consequently, it is not known whether mice NK cells have their CD56bright t and CD56dim counterparts; thus, many NK cell studies are limited to those immunological factors that are conserved between the species [14].

Apart from distinguishing predominant populations based on surface expression of CD56 and CD16, other NK cell subsets have been identified using either flow cytometry or single-cell RNA sequencing. Both methods report unexpected diversity and highlight the need for further studies on the possible implications of newly identified subpopulations [8,15].

### 1.2. Development of NK Cells

NK cells derive from multipotent haematopoietic stem cells that are present in the liver and bone marrow of the foetus. Haematopoietic stem cells (CD34+,CD133+,CD244+) then give rise to lymphoid progenitor cells (CD45RA+) that progress to committed precursors for B cells, T cells, and NK cells [14,16]. NK cells share several properties with adaptive immune system lymphocytes, given the fact that they all derive from the same lymphoid progenitor cell. During their development, NK cells start to express CD122, a well-known marker of NK cell lineage precursors that, together with IL-15Rα (CD215) subunit and common g chain (CD132), makes them sensitive to IL-15 stimulation. IL-15 signalling finally drives the cell maturation into NK cell lineage (CD3-CD56+) [17]. Further differentiation takes place in various tissues, including the spleen, liver, secondary lymphoid organs, thymus, gut, and uterus [14,18,19].

Two models have been used to describe the development of NK cells in humans: linear and nonlinear. In the linear model, NK cells exclusively develop from lymphoid progenitor (LP) cells, as was traditionally accepted. The model postulates a chronological differentiation where CD56bright t NK cells further differentiate into CD56dim populations. This suggests that CD56bright t cells are at the early maturation stage. The hypothesis is supported by in vitro experiments which show that under specific culture conditions, CD56bright t NK cells can convert into a CD56dim population [20].

However, the recent evidence challenged the traditional linear model, indicating that both LPs and myeloid progenitors (MP) can give rise to mature NK cells [21]. Regardless of which model of NK cell development we accept, the studies on this topic clearly show how NK cell phenotype and function can be affected by culture conditions, including intercellular interactions and cytokine milieu. This knowledge is of great importance when designing new NK-cell-based therapeutical approaches [22].

Stages of human NK cell differentiation in vivo remain unknown. It is still not clear if NK cells expressing different receptor phenotypes, located in different tissues, derive from functionally distinct lineage precursors, and/or whether they differ because they were simply affected by tissue-specific factors.

## 2. NK Cell Receptors

NK cells bare a set of inhibitory and activating receptors. Unlike T cell receptors (TCR), NK cell receptors are germline encoded, meaning that they do not undergo gene rearrangement. The balance between activation and inhibition is maintained by ligand/receptor interactions. Healthy cells express self-major histocompatibility complex (MHC) class I molecules on their surface and a vastly small to a non-existent amount of activating ligands. Then, their MHC class I molecules ligate to the killer immunoglobulin-like family (KIR) of inhibitory receptors on NK cells preventing NK cell-mediated lysis [23]. Loss or alterations in MHC class I molecules resulting from infection with an intracellular pathogen, stress, or neoplastic transformation, activates NK cell cytotoxicity. Tumour or virus-infected cells often lose or downregulate MHC class I molecules; thus, they are potential targets for NK cell lysis [24,25,26]. However, in many human tumours, loss of MHC class I molecules does not activate NK cells. Various mechanisms can be involved, including upregulation of inhibitory ligands and non-classical MHC class I molecules HLA-E and HLA-G on the tumour that suppress NK cell cytotoxicity and induce tumour tolerance [27,28,29].

### 2.1. NK Cells Inhibitory Receptors and Their Ligands in Tumour

There are two different families of MHC-specific receptors that have been defined in human NK cells: KIRs and CD94/NKG2 (lectin-like heterodimers) that recognise self-classical and non-classical MHC class I molecules, respectively [30,31,32]. Among KIR and CD94/NKG2 families, both inhibitory and activating receptors have been identified [33]. NK cells do not only respond to the loss of MHC class I, but they also show a degree of selectivity to peptides presented on MHC class I [34]. While the inhibitory receptors induce cell tolerance during recognition of self-MHC class I molecules presenting self-peptides, the activating receptors induce NK cell activation when self-MHC class I molecules are laden with foreign or altered peptides [33,35]. This mechanism enables NK cells to discriminate autologous healthy cells from neoplastic cells. Apart from KIRs and CD94/NKG2 receptors, NK cell responses can be tuned by several other surface molecules, including T cell immunoreceptor with Ig and ITIM domains (TIGIT), programmed cell death protein 1 (PD-1), lymphocyte activation gene 3 (LAG-3), and T cell immunoglobulin and mucin domain-containing protein 3 (Tim-3) [35,36]. Considering the complex regulation of NK cell responses, we will focus only on selected inhibitory receptors that are known to suppress NK cells functions in tumour, such as KIR2DL1–3, KIR3DL1–2, NKG2A/CD94, TIGIT, PD-1, LAG-3, and Tim-3 (Figure 1) [37,38,39,40,41].

MHC is found in many animals, and a human version of it is called the human leukocyte antigen complex (HLA). HLA-specific inhibitory receptors of NK cells bind and recognise HLA class I molecules (HLA-I) of self and healthy cells. This mechanism prevents NK cell activation and induction of autoimmune responses. Thus, the loss of HLA-I or change in their structure makes the cells vulnerable to NK cell lysis [42]. This natural mechanism is of great importance for eliminating tumour cells that downregulate the expression of HLA-I to avoid recognition from cytotoxic CD8+ T-cells (Tc). As expected, KIRs and NKG2 receptors recognise certain epitopes expressed by self-HLA molecules.

For example, KIR3DL2 was found to recognise HLA-A3 and HLA-A11, while KIR3DL1 recognises certain HLA-B and HLA-A allotypes, notably those expressing Bw4 epitope [43,44,45,46,47]. KIR2DLs specificity is largely determined by the amino acid at position 80 in HLA-C molecules. KIR2DL2 and KIR2DL3 recognise group 1 HLA-C (HLA-C1) allotypes that have asparagine residue at position 80, whereas group 2 HLA-C (HLA-C2) allotypes that have lysine at this position confer recognition via KIR2DL1 [47]. Interestingly, NK cells from one individual show high diversity in the expression of HLA-specific receptors [30,36,48]. A signalling blockade via these receptors on NK cells leads to increased cytotoxicity of NK cells [49] Therefore, targeting HLA-I inhibitory receptors is a novel approach in tumour immunotherapy [49,50].

Notably, and in the context of their activating role, the KIR3DL2 inhibitory functional receptor can also bind unmethylated CpG-oligodeoxynucleotides (ODNs) on the cell surface and translocate them to the endosome. Unmethylated CpG-ODNs belong to pathogen-associated molecular patterns (PAMPs) that are unique for the microorganisms but not present in the human body. So after translocation, they are loaded onto toll-like receptors 9 (TLR-9)—a fundamental set of receptors expressed by cells of innate immunity that bind viral and bacterial DNA, but are also associated with infections and cancer [51,52]. KIRs not only play a prominent role in self and non-self-HLA discrimination but, as proposed recently, they are also involved in indirect recognition and response to pathogens or to host cell death [53]. Notably, CpG-ODN binds to KIR3DL2 and induces internalisation of this HLA-I specific inhibitory receptor [54]. This suggests that CpG-ODN may increase NK cell cytotoxicity against HLA-I+ tumours by lowering the density of KIR3DL2 inhibitory receptors on the cell surface.

NKG2A/CD94 heterodimer recognises non-classical HLA molecules and plays a key role in the tolerance of HLA-E+ tumours. Generating NK cells that do not express NKG2A receptor was found to increase their cytotoxicity towards HLA-E-bearing tumours [38]. These observations indicate the potential therapeutic approach based on KIR and NKG2 receptor blockade [55].

T cell immunoglobulin and ITIM domain (TIGIT), also known as WUCAM, Vstm3, and VSIG9, is a recently discovered co-inhibitory receptor classified as a member of the poliovirus receptor (PVR)/nectin family. TIGIT is primarily expressed in T cells and NK cells [56]. In addition, it has been suggested as a marker of natural thymus-derived regulatory T cells (tTregs) with strong suppressive activity and lineage stability [57]. TIGIT knockout leads to hyperproliferative T cell responses, increases levels of proinflammatory cytokines, and hinders IL-10 production [58]. TIGIT expression on tumour infiltrating lymphocytes (TILs) was reported in a variety of malignancies [56,59] and negatively correlated with patient survival [56]. PVR known as CD155 (Necl5 or Tage4) was identified as a high-affinity receptor for TIGIT, while PVRL3 and CD112 (also known as PVRL2/nectin 2) bound TIGIT with lower affinity [60].

In addition, TIGIT-nectin4 interaction was reported in the tumour. Nectin4 is a protein highly expressed during foetal development in human embryos and the placenta, but it is not present in adult tissues. However, it is upregulated in several tumours (including breast, bladder, lung, and pancreas) and was shown to be associated with poor prognosis. In addition, soluble nectin4 has a prognostic value in tumours [61,62,63,64,65]. Recently Reches et al. reported that nectin4 is also a ligand for TIGIT and their interaction inhibits NK cell function, while anti-nectin4 blocking antibodies induced enhanced tumour killing in vitro and in vivo [66]. The effect of TIGIT signalling on NK cells was also confirmed by other studies. Zhang et al. demonstrated that TIGIT expression on tumour-infiltrating NK cells is associated with their impaired function and tumour growth. Even though TIGIT is expressed on both NK cells and T cells, the effect of TIGIT blockade is excessively dependent on the TIGIT expression on NK cells and restores NK cell cytotoxicity. Consequentially, suppression of tumour growth is observed [67]. Thus, TIGIT arises as a novel target for immunotherapy.

Another inhibitory receptor expressed by both NK and T cells is PD-1. Initially, success of anti-PD-1/PD-L1 immunotherapy was suggested to be associated with T-cell activation. However, it was soon observed that lack of HLA class I molecules expression on tumour cells does not abolish the therapeutic effects of the treatment. Therefore, new studies on NK cells in the context of anti-PD-1/PD-L1 immunotherapy were launched. Large numbers of clinical trials evaluating the effects of PD-1/PD-L1 axis inhibition were designed and showed encouraging results. The therapy improved outcome of advanced metastatic melanoma patients, in some cases resulting in full response and clearance of all visible metastases [68]. Compared with previously used chemotherapies, the antibodies improved patient response and led to better disease control. Thus, not surprisingly PD1/ PD-L1 pathway inhibition became the first-line treatment in several cancers. A recent report on anti-PD-1 monotherapy in daily clinical practice showed high rate of long-lasting remissions, regardless of the treatment discontinuation [69]. In the case of patients with late-stage non-small cell lung cancer (NSCLC), PD-1 inhibitors combined with chemotherapy radically improved clinical outcomes when compared to chemotherapy alone [70]. Interestingly, anti-PD-L1 mAbs were found to activate PD-L1+ NK cells for efficient killing of PD-L1 negative tumours in vivo. It was proposed that PD-L1 expression by NK cells is followed by upregulation of the p38 signalling pathway that further enhances NK cells’ cytotoxicity [71]. Nevertheless, not all patients benefit from anti-PD-1/PD-L1 therapy. Interestingly a phenomenon called hyperprogression was observed in some patients after implementation of these antibodies. The mechanisms of this adverse effect have yet to be elucidated. However, studies on NK cell responses in the individuals with rapid tumour progression after inhibition of PD-1/PD-L1 axis might contribute to understanding of this phenomenon [72,73].

Another potential marker of immune suppression in NK cells is the T cell immunoglobulin domain and mucin domain 3 (Tim-3). Ligands that bind Tim-3 include carcinoembryonic-antigen-related cell-adhesion molecule (CEACAM-1), whose overexpression on tumour cells might lead to unrecognition by NK cells. The second high-mobility group box 1 (HMGB1) plays an important role in haematological malignancies [74]. The third most studied ligand for Tim-3 receptor is Galectin-9, which is known to participate in NK cells regulation during early pregnancy [75]. Data coming from patients suffering from lung adenocarcinoma, melanoma, and oesophageal cancer, respectively, associate Tim-3+ NK cells expression with poor prognosis. In all the studies, Tim-3 blockade resulted in increased activity of NK cells [40,76,77].

The expression of LAG-3 on NK cells, as well as an exact mechanism of its inhibition of NK cells, is unknown, but there is supporting evidence that blocking LAG-3 on NK cells increases production of cytokines such as IFN-γ and TNF-α [78]. Combined LAG-3 and PD-1/PD-L1 blocking is extensively studied in clinical trials as one of the most promising approaches in cancer immunotherapy. The approach was reported to double progression-free survival compared with anti-PD-1 monotherapy in patients with previously untreated melanoma [79].

Similar reports have recently shown that CD96 also inhibits cytokine production of NK cells, possibly by competitively binding with DNAM-1 for CD155, making it a potentially interesting target for tumour immunotherapy [80].

### 2.2. NK Cells Activation Receptors and Their Ligands in Tumour

Natural killer cells activation receptors play a significant role in immunosurveillance and activation of NK cells. They include members of different family receptor groups, but we will focus on those receptors that are responsible for immunosurveillance of tumour-like NKG2D, CD16, natural cytotoxicity receptors (NCR), DNAX accessory molecule-1 (DNAM-1), CD224/natural killer cell receptor 2B4, NK-T-B antigen (NTB-A), CD161/NKRP1A, and CD319/CS1 (Figure 1) [81,82,83,84].

The most widely explored activation receptor in NK cells is NKG2D. In humans, NKG2D binds HLA-class I restricted proteins A/B (MICA/B) and UL16-binding protein (ULBP1–6). The expression of ULBPs was reported as high in haematological malignancies, such as leukaemia, and low in solid tumours [85]. These data correspond with better efficacies of NK-cell-based immunotherapies in haematological malignancies versus solid tumours. Interestingly, the expression of MICA/B was reported to be low on the surface of tumour cells [86], while elevated levels of soluble MICA/B were present in the sera of patients with different malignancies [87,88]. Shedding of MICA/B facilitates tumour proliferation, as loss of activating ligands on a tumour cell allows immune escape. In support of this, higher levels of soluble MICA/B in sera of patients were associated with significantly lower overall survival [89]. Additionally, the release of NKG2D ligands leads to reduced density of the ligand on the surface of the tumour, cells subsequently lowering the possibility of cell lysis [86,90]. Moreover, constant stimulation by soluble ligands leads to negative effects and downregulation of NKG2D receptors on the NK cells surface [86,91].

CD16 antibody-dependent cellular cytotoxicity (ADCC) mediated by NK cells is, in many cases, the key immune mechanism through which therapeutic monoclonal antibodies (mAbs) mediate tumour cell killing [81]. Several studies have reported changes in TME after mAbs therapy. Such changes include an increase in tumour infiltrating NK cells that significantly correlates with better clinical outcomes for the patients [92,93,94]. Further supporting evidence of a positive effect of NK cell infiltration and increase in ADCC is that in the studies where Treg inhibition of NK cell was induced, response to mAbs treatment was significantly lower [95,96]. Persistent exposure and/or chronic engagement of CD16 receptor results in a reduction in its surface expression on NK cells, leading to lower tumour cell lysis via NK cytotoxicity, subsequently leading to lower response to mAb-based tumour therapy [97,98].

As mentioned earlier, the killer immunoglobulin-like receptor family (KIR) in natural killer cells binds HLA class I expressed on healthy cells, therefore recognising healthy from unhealthy cells. HLA-mediated NK cells activation against the tumour is based on either blocking KIR inhibitory receptors expressed on NK cells or selecting NK cell donors that mismatch. NK cells cytotoxic potential against tumour cells was higher in the case of HLA-KIR mismatch when compared to autologous donors [99,100].

From the group of natural cytotoxicity receptors (NCRs), NK cells express NKp44-, NKp46-, and NKp30-activating receptors. NCRs participate in tumour cells recognition by binding ligands such as tumour cell-derived HLA-B-Associated transcript 3 (BAT3), B7-H6, mixed-lineage leukaemia protein-5, or properdin, respectively, expressed either intracellularly, extracellularly, or released as a soluble ligand [101,102,103,104]. It should be mentioned that, in addition to the low expression of activating ligands, tumour cells express inhibitory ligands for NCRs. As such, NK cells in TME present NCR low phenotype and are inactive [105].

DNAX accessory molecule-1 (DNAM-1) is another activation receptor involved in tumour immune surveillance. Known ligands for DNAM-1 are CD155 and CD112. As in the case of other activating receptors ligands, chronic exposure to CD155 leads to reduced DNAM-1 expression, and subsequently, NK-cell-decreased responsiveness and tumour progression, as reported in ovarian cancer, glioblastoma, colorectal carcinoma, lung adenocarcinoma, melanoma, and pancreatic cancer [82,106,107,108,109,110]. Apart from being overexpressed on the surface of tumour cells, levels of soluble CD155 molecules have been shown to be high in the sera of cancer patients [111]. Moreover, CD155 is not only expressed by tumour cells but also by myeloid suppressive cells in TME, subsequently leading to low expression of DNAM-1 on both NK and T cells, resulting in their impaired cytotoxic activity [112]. CD155 is the focus of NK checkpoint inhibitor therapy [106]. Both CD112 and CD155 combined as ligands for DNAM-1 have potential in genetically modified NK cell-based therapy for primary sarcoma in in vitro experiments. Being designed to overexpress DNAM-1 activating receptor, genetically modified NK cells efficiently lysed sarcoma cells, while the NK cell line was ineffective [113].

Another activating receptor 2B4 (CD244) interacts with the CD48 ligand that activates NK cells and results in the release of perforin and granzymes [114,115,116]. When human NK cells were stimulated by anti-CD244 mAb, their cytotoxic ability against tumour cells significantly improved. Interestingly, anti-CD244 stimulation had an antagonistic effect on the proliferation of NK cells induced by IL-2 stimulation [117]. Two more receptors, CRACC and NK-T-B antigen (NTB-A), from the same receptor family of signalling lymphocyte activation molecule (SLAM), are also potent stimulators of NK cytotoxicity [83,118].

CS1-activating receptor on NK cells was mostly studied in the context of multiple myeloma [119]. After showing excessive activation of NK cells by mAb elotuzumab via CS1 receptor in clinical trials, it was approved by the FDA in 2015 for the treatment of relapsed or refractory multiple myeloma [120,121].

Blocking of Lectin-like transcript 1 (LLT1) on tumour cells, a ligand of CD161/NKRP1A natural killer cell receptor, resulted in activating lysis against the most invasive triple-negative breast cancer [84].

### 2.3. Toll-like Receptors Based Activation of NK Cells in Tumour

NK cells also express Toll-like receptors that help them recognise and respond to microbial-associated and pathogen-associated molecular patterns (PAMP). They are expressed either on the cell surface (TLR1,2,4,5,6) or on the endosome (TLR3,7,8,9). Though TLR agonists can potentiate NK cell cytotoxicity and cytokine production, their complete activation is still dependent on the cellular microenvironment (as the presence of other immune cell subsets) [122]. A recent review discussed TLR agonist’s current and potential use in boosting NK cell effector function in tumour immunotherapy. It highlights the potential of using TLR agonists as adjuvants for the treatment of breast cancer and melanoma via TLR3 stimulation using viral dsRNA. Interestingly, the effect of TLR7 stimulation via R848 resulted in increased mAbs treatment efficacy in Obinutuzumab treatment of lymphoma, and TLR2 agonist polysaccharide kestin showed promising results as a novel method of treatment of HER2+ breast cancer in combination with Trastuzumab [123].

### 2.4. Balance between Activating and Inhibiting Receptors

Depending on the receptor repertoire, different NK cell subsets will respond to tumour cells in a specific way. Cytotoxic subsets will be activated by stress ligands expressed on the surface of tumour cells that are not present on the surface of healthy cells. Apart from direct cytotoxicity toward tumour cells, certain subsets recruit and promote antitumour immunity of other immune cell subsets through secretion of IFN-γ and chemokines. For example, NK cells that produce cytokines enhance the maturation and activation of dendritic cells, subsequently affecting antigen presentation and T cell-mediated antitumour effect [124].

Cytotoxic T-cells can serially lyse target cells [125]. This means that one cell can repetitively degranulate (release of cytotoxic granules, e.g., perforin and granzymes). Tumour cells lose or downregulate the expression of MHC class I molecules, and therefore evade serial killing via CD8+ T cells [126]. Having a set of cells able to continue tumour cell lysis serially, regardless of MHC-class I downregulation, brings NK cells to the centre of attention. Recent studies show that only a minority of NK cells can serially lyse target cells. The capability of some NK cells to repeatedly create immunological synapses and establish contact with target cells is worth further exploration [127]. Cells with such features are a promising tool, and future studies will focus on the identification of such subpopulations.

In addition to cytolytic degranulation, NK cells can lyse tumour cells by engaging the death receptors that mediate apoptosis. This process is slower than perforin and granzyme-mediated killing. Surface death receptor FasL engages with the CD95 receptor on the target cells and subsequently activates the apoptotic pathway. When compared, this mechanism based on apoptosis activation through death receptor FasL is an elaborate process that involves many steps, unlike granzyme-based apoptosis, which is induced in a fast and dynamic manner [128].

Another death-mediated receptor, TRAIL, is expressed on a variety of immune cells, from cytotoxic effectors such as T-cells and NK cells to monocytes and dendritic cells. Even though resting NK cells do not express detectable levels of TRAIL on their surface, its expression can be induced upon activation, as confirmed in vitro [129]. In acute lymphoblastic leukaemia, the resistance of tumour cells to NK cell-mediated killing was overcome by inducing TRAIL expression on the surface of NK cells [129]. In the future, it will be important to distinguish which subset of NK cells uses which cytotoxic mechanism to harness their full antitumour potential [127] (Figure 2).

## 3. Changes in NK Cells Due to Ageing and Cell Senescence

Ageing is a physiological process affecting single cells, organs, or an entire organism. It manifests as a decline in biological functions, lower regeneration and proliferation potential, and impaired adaptation to stress factors. NK cell immunosurveillance is crucial for elimination of malignantly transformed tumour cells. Therefore, NK cell dysfunction directly increases the risk of tumour development and progression. It was revealed more than 20 years ago that low numbers of NK cells are associated with high risk of mortality in elderly individuals [130], while well-preserved cytolytic activity of NK-cells (comparable to that in healthy young individuals) is a hallmark of the immune system of centenarians [131,132]. Nevertheless, in the majority of individuals, ageing alters NK cell function and population distribution. The continuous increase in CD56dim and decrease in CD56bright t populations has been reported in the elderly [133,134]. Even though the CD56dim subpopulation shows higher cytolytic activity than the CD56bright t subset, their cytotoxicity and cytokine production (e.g., IFN-γ and TNF-α,) becomes impaired with age. Previous studies revealed that NK cells from the elderly have significantly lower expression of activating receptors NKp30 and NKp46 than the cells from younger individuals [135,136]. Further on, ageing affects perforin production and leads to impaired degranulation [136]. Increased numbers of CD56dim NK cells can be a compensatory mechanism in response to cytolytic dysfunction. Not surprisingly, NK cell senescence correlates with a higher incidence of viral infections and tumour development [137]. Another very interesting aspect is a premature senescence of NK cells triggered by the tumour [138]. Thus, both physiological and premature NK cell senescence seems to be an important factor affecting cancer immunotherapy efficacy and creates the broad field for new research on NK cell biology.

## 4. Diverse Factors in the Tumour Microenvironment (TME) Shift the Balance to Inhibition

Even though NK cells tumour infiltration is positively correlated with the prognosis, there are many distinct reasons why a tumour microenvironment is a hostile place for NK cells (Figure 3). Therefore, even after infiltration, there is small or no effect on tumour regression and the reasons for such impaired NK cells activity are not clear.

### 4.1. Number of NK Cells Infiltrating Tumour and Their Chemokine Receptor Repertoire

NK cells’ role in solid tumours is not well understood, but they are present in the tumour microenvironment, although sometimes in low numbers [139]. The latest studies showed a high number of tumour-infiltrating NK cells is associated with good prognosis in metastatic melanoma [140], breast cancer [141], and prostatic cancer [142].

To reach an eminent level of activation and to adequately perform, NK cells need to migrate to tumour tissue. Different natural killer cell subsets express a variety of chemokine receptors on the cell surface, which determines their tissue tropism. CD56bright CD16-cells were reported to express receptor CCR7 [143] that attracts them to the lymph node. Opposed to that, CD56dimCD16+ cells express chemokine receptors that help them leave the lymph node and recruit to the sites of inflammation, such as CXCR1 [144], CXCR3 [145], CX3CR1 [146], and S1P [147].

In physiological conditions, NK cells in the lymph node are CD56bright, but interestingly, in the case of melanoma with metastasis in lymph nodes, increased concentration of serum levels of CXCL8 (ligand for CXCR1) has been reported. In such cases, NK cells present in lymph nodes in which cancer spread surprisingly expressed CD56dim, CD57+, cytotoxic phenotypes. Thus, CXCL8 could play a role in facilitating cytotoxic NK cells from the blood to the TME [148]. Similarly, high expression of CXCL9 and CXCL10 ligands for CXCR3 in ovarian cancer have been associated with increased lymphocyte infiltration, including NK cells, and doubling overall survival rates [149]. Genetically engineered NK cells with both tumour-specific chimeric antigen receptor (CAR) and induced CXCR4 chemokine expression can improve immunotherapy in solid tumours, partially by increasing tumour infiltration of NK cells [150].

### 4.2. Abnormalities in Vascularity and Low Levels of Oxygen within TME Impair NK Cell Cytotoxic Functions

A known characteristic of tumours is hyperpermeability of tumour blood vessels followed by an increase in pro-angiogenic factors such as vascular endothelial growth factor (VEGF) and platelet-derived growth factor (PDGF). Firstly, due to hyperpermeability, and secondly, due to elevated metabolic activity, TME is deprived of oxygen [151]. NK cells in a hypoxic environment have altered phenotype and strongly impaired cytotoxic functions, some of which include: significantly lower cytotoxicity against tumour cells, lower expression of effector protein granzyme B and IFN-γ, as well as degranulation marker CD107a. The surface expression of activating receptors NKp30, NKp46, and NKG2D is also significantly lowered [152]. Unlike the natural killer receptor group, CD16 receptor expression is unchanged. As mentioned, this receptor is involved in ADCC killing and may induce target-cell lysis, even in hypoxic conditions [153]. Proving the importance of different activation pathways, researchers engineered NK cells that express high-affinity CD16 and IL-2 to subsequently increase ADCC and their activation in hypoxia. Unlike donor-derived NK cells, engineered NK cells obtained their cytotoxic function and were resistant to the negative effects of hypoxia. Engineered NK cells are under evaluation for practical utilisation in clinical trials for different tumour types [154].

### 4.3. Low Surface Expression and/or Shedding of NK Activation Receptor Ligands by Tumour Cells

The reduced number of activating ligands on the surface of tumour cells hampers the activation of NK cells. Tumour cells shed MICA from their surface using ADAM protease, and therefore directly reduce the amount of activation ligand for NKG2D [155]. Moreover, released soluble activating ligands bind to NKG2D, thus preventing the interaction with the target cell and resulting in the downregulation of NKG2D on NK cells. Therapeutic blocking of ADAM protease could slow down tumour progression by reducing immune evasion [156]. Protease inhibitors that would prevent the shedding of activating ligands from the surface of tumour cells could improve the efficacy of NK cell-based immunotherapies [157].

### 4.4. Immunosuppressive Cytokines Present in the TME

Cytokines are soluble proteins that regulate immune cells in various tissues. Additionally, cytokines released by immune cells allow them to communicate and maintain tissue homeostasis. Within TME tissue, homeostasis is additionally regulated by tumour orchestrated cytokine immunosuppression, which is critical for tumour immune evasion. The first of such cytokines is transforming growth factor *β*1 (TGF*β*1) which is produced excessively by tumour cells and results in downregulation of NKp30 [158] and NKG2D [159] NK cells activation receptors. Both receptors are involved in direct cytotoxicity toward tumour cells; therefore, this cytokine directly affects NK recognition of tumour cells. Another mechanism by which TGF*β*1 affects NK cells is the downregulation of the CX3CR1 chemokine receptor. This receptor is present on the surface of the cytotoxic CD56dim population, and its downregulation alters NK cells’ ability to migrate and localise [160]. Apart from directly affecting the cytotoxicity of NK cells, TGF*β*1 was recently reported to alter the expression of five different NKG2D ligands in lung cancer, and consequently altered NK immunosurveillance [161]. Developing the mechanism to prevent TGF*β*1 signalling could help restore NK cell cytotoxic potential.

A high concentration of interleukin 10 (IL-10) is another common phenomenon in tumours. Like TGF*β*1, the IL-10 role is dual. Firstly, it suppresses the expression of activating ligands for NKG2D NK cell receptors, and secondly, it induces the expression of the human antigen-G (HLA-G) molecule on tumour cells [162]. It is worth mentioning that this nonclassical MHC-I molecule is of critical importance for pregnancy tolerance [163]. HLA-G interacts with NK cell killer Ig-like receptors (KIRs), resulting in the inhibition of NK cells [164]. Consistently, HLA-G expression in various malignancies was associated with tumour escaping immunosurveillance and was also involved in the mechanism of metastasis, leading to poor clinical outcomes [165].

Interleukin 6 (IL-6) and interleukin 8 (IL-8) production in squamous cell carcinoma impaired the activity of NK cells via downregulation of NKp30 and NKG2D receptors [166]. Increased levels of IL-6 in the peritoneal fluid were found to suppress the production of cytotoxic granules such as granzyme B and perforin in patients suffering from endometriosis, indicating altered NK cell functions when exposed to elevated concentration of this cytokine [167]. Notably, additional evidence of the impact of impaired NK cell function in inflammation was reported in 2020 via a clinical study of COVID-19 patients where increased levels of IL-6 were correlated with decreased levels of granzyme A+ NK cells [168]. Anti-IL-6R antibody Tocilizumab has clinical application to reduce cytokine storm syndrome that is associated with chimeric antigen receptor T-cell therapy. As recently reported, IL-6 present in the microenvironment lowers the immune response induced by chemotherapy, suggesting that targeting IL-6 may, on one hand, suppress autoimmunity, while on the other it potentiates antitumour response [169].

### 4.5. The Effect of Non-Tumour Cell Types in TME on NK Cells

Macrophages in general are separated into two distinct populations depending on their activation pathway: classically activated (M1) and alternatively activated (M2). Tumour-associated macrophages (TAMs) are macrophages present in TME that show pro-tumourigenic features. They present an M2-like phenotype and produce various immunosuppressive molecules, thus participating in tumour escape. Not only TAMs release TGF*β* and IL-10 (NK cell suppressive cytokines that directly alter NK cell cytotoxicity) but also, upon cell-to-cell contact with M2 macrophages NK cells downregulate two activation markers: CD107a and CD27 [170]. Shifting the macrophages phenotype from M2 to M1, which is an antitumoural-like phenotype, was proven to be a promising target to enhance antitumour immunity. Monoclonal antibody therapy in combination with TME administered IL-21, a cytokine that shifts M2 to M1 phenotype, restores cytotoxic effects of CD8+ T cells and/or NK cells [171].

Another cell type, participating in TME homeostasis, are cancer-associated fibroblasts (CAFs), known for various tumour- and metastasis-promoting effects [172,173,174]. CAFs, present in the tumour environment, resist radiotherapy and promote tumour recurrence in cervical cancer [175]. Additionally, CAFs secrete IL-6, TGF*β,* and prostaglandin-2 (PGE2), directly and indirectly stimulating indoleamine-2,3-dioxygenase (IDO) production, and thus suppressing the antitumour activity of NK cells. IDO is an enzyme that catabolises conversion of tryptophan amino acids to toxic metabolites and the presence of IDO in the lymphocyte environment leads to immunotolerance. When co-cultured with NK cells, tumour-derived CAFs inhibit the expression of NKp30 and NKp44 by PGE2, and when co-cultured with melanoma cells, they induce downregulation of the ligands for NK cell activation receptor NKp44 and DNAM-1 [176]. Hepatic carcinoma-derived CAFs produce IL-6 to induce IDO secretion via the dendritic cells in TME [177]. In the presence of IDO catabolites, NK cells downregulate NKp46 and NKG2D receptors, directly disabling ligand recognition on tumour cells via these receptors [178,179]. CAFs modulate immunotolerance and tumour progression, but lack of understanding of the mechanisms and their heterogeneity makes them a promising, yet difficult target for novel immunotherapies.

Additional suppression of NK cells in TME comes from myeloid-derived suppressor cells (MDSCs). This heterogeneous population is present in high percentages of tumours and negatively correlates with NK cell numbers [180]. Cell-to-cell contact via NKp30 receptor on NK cells with MDSC resulted in their inhibition in the case of patients suffering from hepatocellular carcinoma [181]. TGF*β*1 expressed on MDSCs induced downregulation of NKG2D on the surface of NK cells, as well as production of IFNγ. IFNγ is a cytokine that potentiates adaptive immune response. Therefore, the anergy of NK cells can potentially lead to impaired functions of the adaptive immune system [182]. MDSC-based suppression of the antitumour activity of NK cells does not restrict to cell-to-cell interactions, as MDSCs release soluble ligands that inhibit NK cells CD16 receptor signalling and alter their antibody-dependent cellular cytotoxicity [183]. Various strategies for improving NK cell-mediated immunotherapies by targeting MDSC were recently reviewed by the authors, who encouraged the implementation of combinatory strategies to increase antitumour effects [184].

T regulatory cells’ (T-regs) biological ability to induce tolerance and prevent autoimmunity is a mechanism misused in the tumour microenvironment (TME). A TGF*β*-rich environment has a positive impact on T-reg function, proliferation, and survival. This leads to a high number of T-regs in the TME, and subsequently leads to poor clinical prognosis and overall survival in many tumour types [185]. T-regs reduce some of the beneficial effects of immunotherapies in several ways [186]. After co-culture with T-regs, NK cells have a low level of perforin and decreased production of IFN-y accompanied by downregulation of NKG2D, NKp46, and NKp30 [187]. Altogether, T-reg modulation of NK cells phenotype drives them to anergy. Reversing this anergy could be achieved by either depleting T-regs from the tumour environment or inhibiting TGF*β* signalling on NK cells. Since T-reg are not the only producers of this immunosuppressive cytokine, we hypothesise that combinatory strategy could yield greater results on the activation status of NK cells.

B regulatory cells (B-regs), the immunosuppressive hand of B lymphocytes, produce elevated levels of IL-10 and TGFβ [188]. Even though T-regs are more frequently studied for their immunosuppressive properties, B-regs are equally responsible for impaired functions of NK cells. Recently, it was demonstrated that B-regs are present in the bone marrow of patients suffering from multiple myeloma (TME) and not in peripheral blood. Furthermore, when co-cultured with NK cells, B-regs derived from TME completely inhibited the ADCC of NK cells when stimulated with therapeutical antibodies [189]. This indicates the necessity of further exploration of the effects B-regs have in antibody-dependent immunotherapies, as their ability to impair NK cell function is strong. Their novel function was recently identified, and they may not have been the focus of the research due to the inability to identify their phenotype or due to their low numbers present within TME.

It is also worth mentioning that dendritic cells are known for their ability to enhance NK cell activation via the production of IL-12, IL-15, and IL-18 cytokines. These cytokines are crucial for complete NK cell activation, proliferation, survival, and response to infection [190], and exactly this cytokine priming mechanism found its clinical application in tumour immunotherapy as cytokine-induced-memory-like NK cells [191]. The tumour microenvironment can indeed turn a friend into a foe, as shown in the case of human lung adenocarcinoma, where dendritic cells were reported to produce elevated levels of TGF*β* upon encounter with tumour cells compared to cells that were not in contact [192]. DC-based vaccines, which would fully elicit an innate and adaptive antitumour immune response, could overcome the negative effects of TGF*β* on NK cells.

### 4.6. Immunosuppressive Metabolites Released by Tumour Cells

Prostaglandin E2 (PGE2) is a metabolite of cyclooxygenase-2 (COX2) enzyme associated with multiple tumour types such as breast, lung, colorectal, and pancreatic tumours, where they contribute to immune system dysfunction [193,194]. Tumour cells produce PGE2 and subsequently inhibit NK cell differentiation and maturation, as well as reducing the level of activating receptors such as NKp44 and NKp30 together with death receptor ligand TRAIL [195]. This directly limits NK cell cytotoxic potential for tumour cells expressing TRAIL-R.

A hypoxic environment within the tumour leads to the accumulation of adenosine, another critical metabolite that suppresses NK cells by affecting their maturation, altering the expression of NKG2D as well as inhibiting TNF α release [196,197]. A solid tumour environment is especially associated with low oxygen concentration, cellular stress, and apoptosis, which results in the release of adenosine [198]. Various solid tumours have high adenosine concentrations that contribute to impaired functions of NK cells. High expression of adenosine A3 receptor was detected in cytotoxic NK cells infiltrating hepatocellular carcinoma, and similar findings supported this data in thyroid, gastric, and colorectal, but not lung cancer [199]. Because of this, and previous evidence of adenosine-based NK cells inhibition, this receptor is worth further exploration as a blocking target might lead to therapeutic benefits in certain tumour types.

Survivin is an anti-apoptotic protein that is usually found within the tumour cells of various organelles, but it is also released and present in TME. Survivin is involved in the regulation of apoptosis as well as cell division in tumour cells, and its high expression is associated with tumour metastasis in melanoma and breast cancer [200,201]. Co-culture where NK cells were directly exposed to survivin resulted in a reduction in perforin and granzyme B, as well as lower production of TNFα and IFN-γ [202].

Extracellular vesicles (EV), the smallest of which are exosomes, are used by tumour cells to communicate with distant cells and tissues. These vesicles contain different molecules that support tumour cell growth by shaping the tumour microenvironment, immune system activation/anergy, intracellular signalling, and angiogenesis [203,204,205]. Tumour-derived EVs were reported to escape perforin-mediated cytotoxicity [206]. Exosomes contain tumour-derived nucleic acids, lipids, proteins, and carbohydrates, making them suitable for early detection and monitoring of disease progression. Collecting various body fluids and analysis of EVs is an example of a liquid biopsy that is a simple, safe and non-invasive method compared with surgical biopsy [207]. Depending on the content of the granules, authors reported various effects on NK cells: (i) reduced migration to the TME [208]; (ii) decreased number and viability [209]; (iii) prevention of cytotoxicity [210]; (iv) lower cytokine production [210]. Tumour-derived exosomes contain molecules similar to those present in TME responsible for immunosuppression such as TGF*β* [204], FasL [211], PD-L1 [212], adenosine [206], survivin [213], and ligands for NKG2D [214], all with high potential to alter NK cell antitumour capabilities.

## 5. Future Perspectives

Mechanisms of immune evasion, developed by solid tumours, present a challenge for NK-cell-based immunotherapies. Although researchers agree that cancer and its microenvironment significantly impair the efficacy of NK cell-based therapies, the previous clinical trials did not address this issue sufficiently, as NK cells were usually implemented as monotherapy [191,215,216,217,218].

NK cell ex vivo expansion results in upregulation of NK cell activation receptors. The true challenge for NK cells is to reach the TME, infiltrate the tumour and remain in their activated state. The relationship between activating and inhibitory signals present in the tumour microenvironment and the effects of immunosuppressive metabolites and tumour promoting cell subsets on NK cells needs further exploration. Such experiments would help us to develop activation and expansion techniques that would generate NK cells resistant to inhibition within the TME. Three-dimensional (3D) tumour cultures are a promising tool for the reliable assessment of NK cell cytotoxicity [2]. Three-dimensional matrices provide a great environment for cancer and cancer derived fibroblast cocultures as well as a great tool for testing the capability of immune cells for tumour infiltration. Additionally, experimental settings mimicking hypoxia closer resemble the TME [154]. The design of highly potent antitumour NK cells in vitro does not always reach expected clinical efficacy, as NK cell tumour infiltration and performance are not solely dependent on expression of activation receptors. In addition, the activation status of any immune cell, including NK cells, is never constant, but is highly variable and affected by the environment.

Solid tumours are often heterogeneous and tumour type, stage, and surface protein expression could greatly affect NK cell responses. Immune checkpoint inhibitors were shown to improve antitumour responses of NK cells. However, NK cells express more than just one inhibitory receptor, so targeting more than one immune checkpoint inhibitor could be beneficial.

Recent studies have demonstrated that a combination of irreversible electroporation (IRE) and NK cell-based immunotherapies has the potential to improve the survival of patients with various solid tumours. IRE is a non-invasive and non-thermal ablation technique that allows the destruction of the tumour microenvironment with minimal damage to associated tissue. Such a method, in combination with NK cell adoptive transfer, increased the number of patients responding to therapy from 68.2% to 88.9% [219].

Some fields of NK cell biology need further exploration, as this knowledge can greatly improve immunotherapy design. One example is the capability of NK cells to lyse tumour cells in a serial manner [220]. In addition, little is known about crosstalk between the most potent effectors of innate and adaptive immunity, such as NK cells and cytotoxic T lymphocytes (CTLs). Addressing all these issues will lead to significant improvement of NK cell production technology. This will probably result in reduction in the therapeutic dose of NK cells required for efficient treatment. Nevertheless, no matter how potent the NK cells and how many we will be able to design and produce, the tumour will not be eradicated completely, unless we elaborate novel strategies protecting these NK cells from tumour immunosuppression.

## 6. Conclusions

Knowing that NK cell cytotoxic response is based on the balance of their activation and inhibitory receptors with a variety of inhibitory signals in the tumour environment, it is hard to shift that balance towards activation. We know that efficient antitumour immune systems response must consist of both innate and adaptive responsiveness and activation. Today’s immunotherapies are not always sufficient, especially those targeting NK cells alone. We should think about NK cells as cells that are majorly affected by the environment they are in, and for their maximum cytotoxic effect and activation, we need to additionally target creation of an environment in which NK cells will be able to perform.

## Figures and Tables

**Figure 1 ijms-23-07030-f001:**
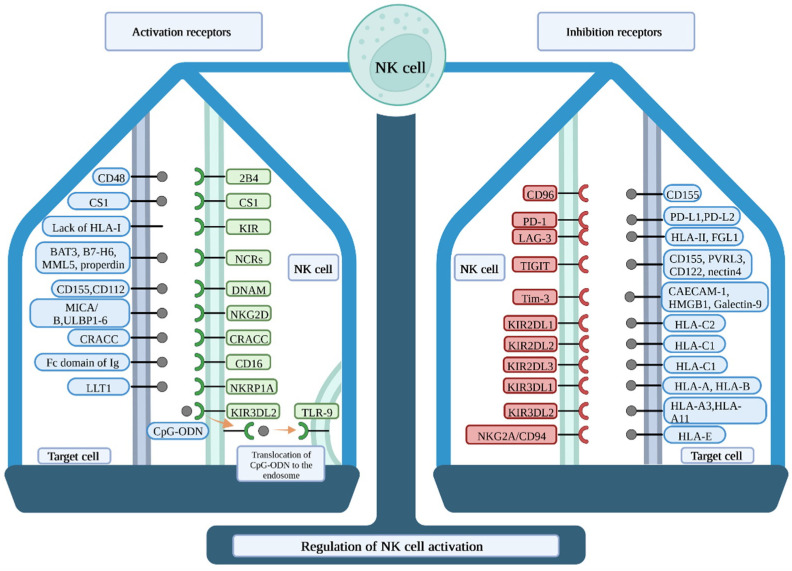
Regulation of NK cell activation. NK cells express a repertoire of activating and inhibitory receptors that bind ligands on the target cell surface. Upon ligation, NK cells can either inhibit or activate their cytotoxic functions. The receptors are potential targets for immunotherapies, as they switch NK cells from the resting to activated state and vice versa. 2B4—CD224/natural killer cell receptor 2B4; BAT-3—tumour cell-derived HLA-B-associated transcript 3; CpG-ODN—CpG-oligodeoxynucleotides; CRACC—CD2-like receptor-activating cytotoxic cell; DNAM—DNAX accessory molecule-1; FGL1—fibrinogen-like 1; HMGB1—high mobility group box 1; HLA—human leukocyte antigen; Ig—immunoglobulin; KIR—killer Ig-Like receptors; LAG3—lymphocyte activating 3; LLT1—lectin-like transcript 1; MICA/B—MHC class I chain-related protein A and B; NCRs—natural cytotoxicity receptors; NKRP1A—killer cell lectin-like receptor subfamily; PD-1—programmed cell death protein 1; PD-L1/2—programmed death-ligand 1 and 2; PVLR3—poliovirus-like receptor 3; TLR-9—toll-like receptor 9; TIGIT—T cell immunoreceptor with Ig and ITIM domains; TIM-3—T-cell immunoglobulin mucin-3; ULBP1–6—UL16 binding protein 1–6.

**Figure 2 ijms-23-07030-f002:**
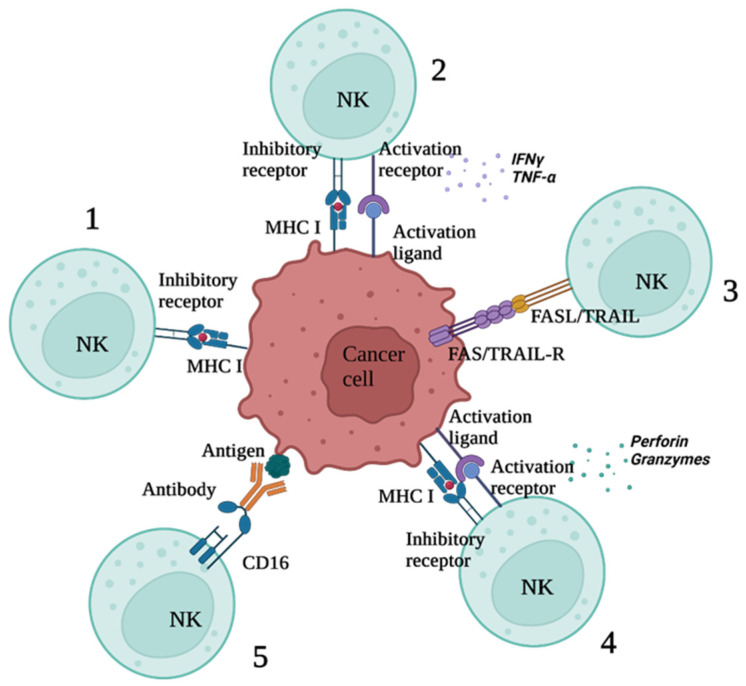
NK cells respond to tumour cells based on their receptor repertoire: (1) ligation of NK cell KIR-inhibitory receptor with MHC-I induces inhibitory signal, lack of which leads to activation; (2) immunoregulatory NK cell subsets, upon binding activation ligands that overcome inhibitory signals, start secreting IFNγ and TNF-α, which promotes maturation and activation of other lymphocytes; (3) NK cell subsets that express death ligands such as TRAIL and FasL will mediate apoptosis in tumour cells expressing adequate receptors; (4) cytotoxic NK cells reaching activation threshold release granules containing membrane-perforating and apoptosis-inducing molecules, such as perforin and granzyme B (degranulation); (5) NK cells expressing CD16 engage in ADCC by lysing tumour cells opsonised by antibodies. ADCC—antibody-dependent cellular cytotoxicity; IFN—γ-interferon-gamma; MHC—major histocompatibility complex; NK cell—natural killer cells; TNFα—tumour necrosis factor-alpha; TRAIL—TNF-related apoptosis-inducing ligand.

**Figure 3 ijms-23-07030-f003:**
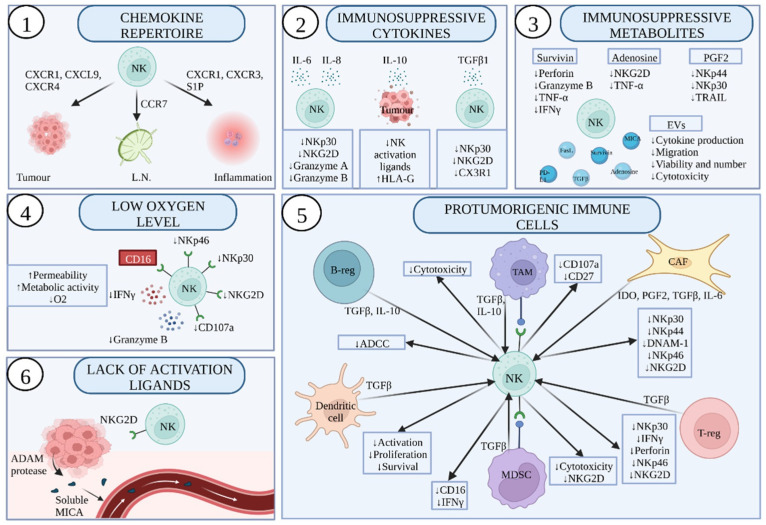
Factors that inhibit NK cells performance in the tumour microenvironment (TME): (1) chemokines and chemokine receptors on NK cells direct them to lymph node (L.N.), tumour or inflammation site; (2) immunosuppressive cytokines lower the expression of activating receptors on NK cells and amount of cytolytic granules, while on tumour cells, they lower ligands for NK activation, and increase ligands for NK inhibitory receptors; (3) immunosuppressive metabolites, soluble or released as extracellular vesicles (EVs), alter NK cell functions; (4) low oxygen levels in TME lower the expression of NK cells activating receptors, but not CD16 receptor responsible for ADCC; (5) protumourogenic immune cells release soluble molecules that alter NK cell antitumour activity, and in case of cell-to-cell interactions with MDSCs and TAMs with NK cells, NK cells function alters; (6) tumour cells reduce the expression of NK cells activating ligands by the activity of ADAM protease. ADAM—a disintegrin and metalloproteases; B-reg—B regulatory cell; CAF—cancer-associated fibroblast; CCR7—C-C chemokine receptor type 7; CXCL9—chemokine ligand 9; CXCR1—C-X-C motif chemokine receptor 1; CXCR3—C-X-C motif chemokine receptor 3; CXCR4—C-X-C chemokine receptor type 4; EVs—extracellular vesicles; FasL—Fas receptor ligand; IFN-γ—interferon-gamma; IL—interleukin; MDSC—myeloid-derived suppressor cells; MICA—MHC class I chain-related protein A; PD-L1—programmed death-ligand 1; PGF—prostaglandin F; S1P—sphingosine-1-phosphate receptors; TAM—tumour-associated macrophage; TGF-β—transforming growth factor-beta; TNF—tumour necrosis factor-alpha; TRAIL—tumour-necrosis-factor-related apoptosis-inducing ligand; T-reg—T regulatory cell.

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
