# Peer review of "What Inhibits Natural Killers’ Performance in Tumour"

_ijms, 2022, doi:10.3390/ijms23137030_

Round 1

Reviewer 1 Report

Please, see attached a document with my comments.

Reviewer 2 Report

The review article "What Inhibits Natural Killers' Performance in Tumour" aims at providing the current understanding of the inhibitory signals towards their cytotoxic potential. The authors have done a good job in providing such a detailed review in a simple form that reads well. 

My only minor suggestion would be to include a section about how aging affects the Nk cell functions. Aging itself is associated with a change in Nk cell phenotype and function. Given the aim of the review, adding the aging aspect would help the scientific community to know how a natural event like aging itself affects the cytotoxic potential of these cells. 

Reviewer 3 Report

In this article, the authors provide an exhaustive review of mechanisms mediating the inhibition of NK cell activity. A general comment in this work is that the authors could include examples of targeting NK cells in the clinic using immunotherapy as a primary or secondary outcome, if the authors find it fitting. Furthermore, the introduction section should be modified to include a general overview of the manuscript, presenting the specific aims of this work. Importantly, it would greatly benefit the article if the authors presented the information in a more critical manner, also adding a future perspectives section that would present the challenges in targeting NK cells in the tumor microenvironment and how we could possibly overcome them.
